# Neuromuscular Fatigue at Task Failure and During Immediate Recovery after Isometric Knee Extension Trials

**DOI:** 10.3390/sports6040156

**Published:** 2018-11-28

**Authors:** Christian Froyd, Fernando G. Beltrami, Timothy D. Noakes

**Affiliations:** 1Faculty of Education, Arts and Sport, Western Norway University of Applied Sciences, 6856 Sogndal, Norway; 2Division of Exercise Science and Sports Medicine, Department of Human Biology, University of Cape Town, Newlands 7725, South Africa; timothy.noakes@uct.ac.za; 3Exercise Physiology Lab, Institute of Human Movement Sciences and Sport, ETH Zurich 8057, Zurich, Switzerland; fernando.beltrami@hest.ethz.ch

**Keywords:** maximal voluntary contraction, peripheral fatigue, neuromuscular activation, femoral nerve electrical stimulation, critical peripheral fatigue threshold, electromyography

## Abstract

We asked whether the level of peripheral fatigue would differ when three consecutive exercise trials were completed to task failure, and whether there would be delayed recovery in maximal voluntary contraction (MVC) force, neuromuscular activation and peripheral fatigue following task failure. Ten trained sport students performed three consecutive knee extension isometric trials (T1, T2, T3) to task failure without breaks between trials. T1 and T2 consisted of repeated 5-s contractions followed by 5-s rests. In T1, contractions were performed at a target force at 60% pre-exercise MVC. In T2, all contractions were MVCs, and task failure occurred at 50% MVC. T3 was a sustained MVC performed until force fell below 15% MVC. Evoked force responses to supramaximal electrical femoral nerve stimulation were recorded to assess peripheral fatigue. Electromyography signals were normalized to an M-wave amplitude to assess neuromuscular activation. Lower levels of evoked peak forces were observed at T3 compared with T2 and T1. Within 5 s of task failure in T3, MVC force and neuromuscular activation recovered substantially without any recovery in evoked peak force. Neuromuscular activation 5–10 s after T3 was unchanged from pre-exercise values, however, evoked peak forces were substantially reduced. These results challenge the existence of a critical peripheral fatigue threshold that reduces neuromuscular activation. Since neuromuscular activation changed independently of any change in evoked peak force, immediate recovery in force production after exercise is due to increased central recruitment and not to peripheral mechanisms.

## 1. Introduction

Although it is accepted that fatigue influences human exercise performance, there is still much to be learned about the exact mechanisms causing this phenomenon [1,2]. Neuromuscular fatigue during exercise is measured as a reduction in maximal voluntary contraction (MVC) force [3] and is believed to result from two different processes termed peripheral and central fatigue. Peripheral fatigue—defined as a reduction in force originating from sites at or distal to the neuromuscular junction [3]—is commonly measured as a reduction in evoked muscle force production in response to electrical supramaximal stimulations delivered to the motor nerve of relaxed muscles [4,5]. Peripheral fatigue during exercise can be caused by impaired action potential transmission along the sarcolemma, by alterations in excitation-contraction coupling or in the actin-myosin interaction [3]. Central fatigue is defined as a reduction in the capacity of the central nervous system to recruit motor units to produce a maximal force [3]; it is commonly measured as a reduction in the level of maximal voluntary activation with use of the interpolated twitch technique [5].

The study of extent and mechanisms by which neuromuscular fatigue contributes to the regulation and limitation of exercise performance continues to attract significant research efforts [1,6,7,8]. One proposal is that the development of peripheral fatigue plays a key role in the regulation of central motor drive (voluntary descending drive from the primary motor cortex), also known as neuromuscular activation, which is measured as the amplitude of the electromyography (EMG) signal during exercise tasks. This theory holds that inhibitory feedback from group III and IV afferents from locomotor muscles to the central nervous system [9,10] limit neuromuscular activation specifically to prevent the development of peripheral fatigue above some critical threshold level, beyond which the level of sensory input becomes intolerable [3].

In this way, the development of peripheral fatigue would reduce neuromuscular activation, and thus, determine the exercise intensity during time trials [11] or the onset of task failure during constant load exercise [10]. The reason is that by design in this paradigm, power output cannot be modified. This model predicts that the level of peripheral fatigue that develops during any form of exercise is task-specific and must always be the same at task failure—defined as the inability to maintain the target force—since a critical threshold of peripheral fatigue was reached [9,10]. However, some authors dispute the existence of a single critical threshold of peripheral fatigue [8,12,13,14,15,16,17].

Even if this critical threshold does exist, its capacity to regulate or limit endurance performance can be questioned. For example, substantial levels of peripheral fatigue in the quadriceps femoris muscles do not limit an end-spurt in neuromuscular activation in self-paced exercise of 3–40 min duration [14]. Furthermore, when exercise duration is short (~3 min), the significant presence of peripheral fatigue at the end of exercise is not necessarily accompanied by reductions in neuromuscular activation [14]. Additionally, during constant-load intermittent isometric exercise trials to task failure, neuromuscular activation increases simultaneously with increased peripheral fatigue [18,19]. Furthermore, during sustained MVC contractions, reductions in voluntary force are also not necessarily associated with decreases in EMG activity [20]. 

It has also been demonstrated that voluntary force starts to recover immediately after sustained MVCs [21]; sustained isometric exercise of the knee extensors [22,23] and other muscle groups [24,25] to task failure; and after cycling to task failure [8,26]. Additionally, a simultaneous recovery in post-exercise voluntary force and in neuromuscular activation has been found immediately [23], and within 10 [26] and 15–20 s [24] after task failure. Only the study by Neyroud et al. [23] evaluated simultaneous changes in MVC force, RMS (root mean square) and evoked peak force (potentiated twitches) immediately after task failure. However, it remains to be established exactly how changes in these factors interact immediately after task failure of a sustained MVC of the knee extensors, and if changes in neuromuscular activation are independent on recovery in peripheral fatigue, as has been hypothesized [27]. In fact, Morales-Alamo et al. [27] and Torres-Peralta et al. [26] documented a functional reserve in force (recovery in force and EMG immediately after task failure), though without any recovery in muscle metabolites. Although no measurements of evoked peak force were made, the results of Morales-Alamo et al. [27] led the authors to suggest that task failure during an incremental exercise to exhaustion depends more on central than peripheral mechanisms. 

Therefore, we designed a study using repeated trials of isometric knee extension exercise to task failure on a dynamometer, in which the threshold for task failure was progressively lowered in each subsequent exercise trial. Our goal was to generate as much neuromuscular fatigue as possible. This experimental method allowed us to (i) investigate changes in neuromuscular activation and evoked peak force at task failure between exercise trials and (ii) examine the simultaneous change in neuromuscular activation, MVC force and evoked peak force within 5–10 s after task failure of the final trial, before any significant recovery in peripheral fatigue would have been expected to occur. 

We hypothesized that evoked peak force at task failure would decrease progressively from the first to the final trial. We further hypothesized that MVC force would recover substantially immediately after task failure in the final trial and that MVC force recovery would match recovery in neuromuscular activation, but with little or no recovery in electrically-induced evoked peak force, the measure of peripheral fatigue.

## 2. Materials and Methods

### 2.1. Subjects

Ten trained sport students (five men, mean ± SD age: 24 ± 4 years, body mass: 71 ± 12 kg, height: 176 ± 9 cm) participated in the study. Subjects were well-trained in both endurance and strength exercises (training more than five times a week in total). At the time of testing, none of the subjects had any leg injuries or experienced knee pain during exercise or testing. Subjects were instructed to refrain from high-intensity exercise on the day prior to testing and to refrain from alcohol during the last 24 h before testing. Subjects were also instructed to eat a light meal 2–4 h before arrival at the laboratory. The study was approved by the Regional Ethics Committee in Norway (2011/1634), and the experiments were performed according to the latest (2013) revision of the Declaration of Helsinki. Informed consent was obtained from all individual participants included in the study. Subjects were given a full explanation of the details and rationale of the study and were informed that they were free to withdraw at any time. The possibility that electrical stimulation might cause discomfort was fully explained as was the nature of the risks involved.

### 2.2. Experimental Design

Subjects visited the laboratory on three occasions. During the first visit, subjects were familiarized with the procedures that would be used for assessing neuromuscular function, comprising electrical stimulation and isometric MVC. In addition, subjects were familiarized with the experimental trials, which involved isometric contractions at a fixed intensity until task failure with knee extension of the right leg on the KinCom dynamometer (Kinematic Communicator, Chattecx Corp., Chattanooga, TN, USA). After the familiarization visit, the subjects made two visits to the laboratory 3–5 days apart. On the first experimental visit, subjects performed two trials to task failure separated by eight minutes of recovery; results from these experiments are reported elsewhere [15]. On the second experimental visit, the subjects performed three trials to task failure, with varying degrees of task failure.

The subjects performed three consecutive isometric knee extension trials in which they were asked to continue for as long as possible, that is, until task failure (Figure 1). By using an isokinetic dynamometer, we were able to measure endurance exercise to task failure and to begin assessment of peripheral fatigue within 2 s following completion of the final MVC. During the first trial, the subjects performed consecutive sets of 10 × 5-s isometric contractions with 5-s rest between contractions. The first nine contractions were performed at a target force of 60% pre-exercise MVC, while the 10th contraction in each set was an MVC. Electrical stimulation to assess neuromuscular function was applied in the 5-s break before the next set of contractions began. During all trials, a target line on a computer screen was used to provide visual feedback of the achieved force. Task failure was defined as the moment the subject could no longer maintain the required force (at or above the target line) for at least 4 s for two consecutive contractions. Subjects were informed each time they failed to achieve the required force output. Following the second missed contraction, defined as task failure of the first trial (TF1), subjects were instructed to produce a final 5-s MVC. The MVC was followed directly within 2 s by electrical stimulation. The second trial started immediately after completion of that series of electrical stimulations, that is, 5 s after task failure. Sets of 10 × 5-s MVCs followed by 5-s rests between MVCs were then performed until voluntary force dropped below 50% of the pre-exercise MVC for more than two consecutive contractions. Following the second missed contraction, defined as task failure of the second trial 2 (TF2), subjects were instructed to produce a final 5-s MVC. The MVC was followed directly within 2 s by electrical stimulation. The third trial started immediately after completion of these electrical stimulations, that is, 5 s after task failure. In this stage of the trial, a sustained isometric MVC was performed until the generated force dropped below 15% pre-exercise MVC, which was defined as task failure in the third trial (TF3). TF3 comprised the last 5 s of the third trial and the following electrical stimulation which began within 2 s. Five seconds after the end of the third trial, another 5-s MVC and electrical stimulation was performed. This is defined as TF3-post.

### 2.3. Procedures

On arrival at the laboratory, subjects were secured to the dynamometer. The seat’s backrest was reclined 10 degrees, and the dynamometer’s rotation arm was kept at 90 degrees. Hip angle and right leg knee angle were approximately 110 and 80 degrees, respectively. The warm-up consisted of 5-s isometric contractions followed by 5-s rest at 25%, 50% and 75% MVC force. MVC force from the first experimental visit was used to determine the intensity for the warm-up exercises.

Neuromuscular function assessment consisted of a 5-s MVC followed, within 2 s, by a sequence of electrical stimuli. Femoral nerve electrical stimulation on relaxed muscles consisted of a single stimulus (SS), paired stimuli at 10 Hz (PS10) and paired stimuli at 100 Hz (PS100). The interval between the stimulation techniques was 1.5 s. In addition, PS100 was followed by tetanus (50 stimuli at 100 Hz = 0.5 s) once at baseline and at TF1, TF2 and TF3-post. Pre-exercise neuromuscular function (Figure 1) assessment consisted of three 5-s isometric MVCs, followed by electrical stimulation. Neuromuscular function was also measured after each set during the trials and at task failure. Power Lab (ADInstruments Pty Ltd., Bella Vista, Australia) was used to trigger the electrical stimulation.

### 2.4. Data Collection

A high voltage (maximal voltage 400 V) electrical stimulator (DS7AH, Digitimer, Hertfordshire, UK) was used to deliver square-wave stimuli of 1 ms duration to the femoral nerve via a 10 mm diameter self-adhesive cathode electrode (Skintact; Leonhard Lang GmbH, Innsbruck, Austria) pressed manually by the investigator onto the skin at the femoral triangle. The triangle is a subfacial space which appears as a depression when the thigh is flexed. The anode, a 130 × 80 mm self-adhesive electrode (Cefar-Compex, Scandinavia, Sweden) was applied to the gluteal fold. The stimulation current (range 35–60 mA) was increased by 30% above a plateau in evoked force response of SS to ensure supramaximal stimulation. The stimulation current was kept constant for the same subject for all types of electrical stimulation. These methods were the same as those used in our previous studies [14,15].

Electromyography (EMG) signals from the vastus lateralis (VL) and vastus medialis (VM) of the right leg were recorded via surface electrodes (DE-2.1 single differential surface sensors, distance between muscle site contacts = 10 mm; Delsys Inc., Boston, MA, USA). SENIAM (Surface Electromyoggraphy for the Non-Invasive Assessment of Muscles) [28] recommendations were used for the placement of the sensors on the skin. Sensors were placed in a direction parallel to the general direction of muscle fibers. The reference electrode was placed on the patella bone. EMG signals were sampled at 2000 Hz and amplified (gain = 1000) using Bagnoli-8 (Delsys Inc., Boston, MA, USA), transferred together with simultaneous force and electrical stimulation recordings into Power Lab (ADInstruments Pty Ltd., Bella Vista, Australia) and filtered using a band pass filter with a bandwidth at 15–500 Hz in Lab Chart Pro software, version 7.3.8 (ADInstruments Pty Ltd., Bella Vista, Australia).

Ratings of perceived exertion (RPE), defined as “the conscious sensation of how hard, heavy, and strenuous exercise is” [29], was assessed after every eighth contractions in each set for the first two trials using the 15 points RPE scale [30]. Subjects were asked to rate how hard they were driving their leg during the exercise, but not to include an assessment of pain in their legs.

### 2.5. Data Analysis

The mean of the three successful MVCs prior to the first trial of the second experimental visit was taken as the pre-exercise MVC force. Since the first and second experimental visits were originally conceived as a single study design, pre-exercise MVC for the first experimental visit was used for calculation of the target force for all trials. Pre-exercise MVC force was not significantly different (*p* = 0.115) between the first (547 ± 123 N) and second (571 ± 137 N) experimental visit. MVC force was calculated as the highest average force sustained for 1 s for the 5-s MVC, i.e., 500 ms before and after peak force. MVC force at TF3 was calculated as the average force during the last 5 s of the third trial. The force responses to electrical stimulation are reported as evoked peak force, and a reduction in evoked peak force is a measure of peripheral fatigue. PS10∙PS100^−1^ (evoked peak force for PS10∙PS100^−1^) was calculated as an index of low-frequency fatigue [4].

RMS of the EMG data was calculated for 1 s around peak force for the 5-s MVC, and for the middle 4 s of the first nine contractions of each set. RMS of the EMG data at TF3 was averaged during the last 5 s of the third trial. M-wave peak-to-peak amplitude in response to SS was also assessed. RMS during MVCs was divided by the M-wave peak to peak amplitude of the following SS response to estimate neuromuscular activation (RMS∙M^−1^) [5]. In several studies, RMS∙M^−1^ has been used to calculate neuromuscular activation [14,15,26], whereas RMS has been used to calculate the central motor drive [11].

### 2.6. Statistical Analyses

After checking for the normality of data distribution using the Shaprio-Wilk’s test, one-way repeated-measures ANOVAs with Bonferroni post hoc corrections were used to detect differences over time. Where the assumption of sphericity (Mauchy’s test) was violated, the Greenhouse-Geisser Epsilon correction was applied to the degrees of freedom. A paired samples student’s *t*-test was used for the pairwise comparisons for differences between the two first trials in RPE and set force during the last sets. The statistical significance was defined at *p* < 0.05. All analyses were performed using SPSS version 24 (SPSS, Inc., Chicago, IL, USA). The results are presented as mean ± SD. GraphPad Prism, v7.03 (La Jolla, CA, USA) and Microsoft PowerPoint 2016 (Redmond, WA, USA) were used to create figures.

## 3. Results

Time to task failure was longer (η² = 0.848, *p* < 0.05) during the first trial (15:44 ± 5:31 min) compared with the second trial (10:31 ± 4:14 min). The third trial lasted 0:41 ± 0:19 min and was shorter than the first and second trial (*p* < 0.001). Although the target force achieved during the first trial was predetermined by the protocol, RMS∙M^−1^ for each set increased, and MVC force and evoked peak force decreased progressively (Figure 2). RMS∙M^−1^ for VM during the last set of both the first and second trial, and for VL during the last set of the second trial was lower (*p* < 0.05) than pre-exercise MVC. Force and RMS∙M^−1^ decreased during the third trial (Figure 3), and was lower (*p* < 0.01) during the last 5 s (TF3) compared with TF2 (Table 1). RPE increased progressively during the first trial, from 11.6 ± 1.7 during the first set of contractions to 18.7 ± 1.5 at task failure. RPE was higher (η² = 0.474, *p* < 0.05) at task failure during the second trial (19.7 ± 0.4) than during the first trial.

MVC force was lower (*p* < 0.001) at TF2 compared with TF1 and at TF3 compared with TF2 (Figure 4 and Table 1). MVC RMS∙M^−1^ was lower (*p* < 0.01) at TF3 compared with TF2 (Figure 4 and Table 1). Evoked peak forces for SS, PS10, PS100 and tetanus were lower (*p* < 0.05) at TF3 or TF3-post compared with both TF1 and TF2 (Table 1 and Figure 4). PS10∙PS100^−1^ ratio was lower (*p* < 0.01) at TF2 compared with TF1, and at TF3 compared with TF2 and TF1 (Table 1).

MVC force (Table 1 and Figure 4) recovered rapidly after TF3, increasing twofold (*p* < 0.01) within 5 s (TF3-post). Within 5 s after TF3, RMS∙M^−1^ recovered (*p* < 0.001) so that RMS∙M^−1^ at TF3-post was no longer different from that measured during the pre-exercise MVC (Table 1 and Figure 4). Interestingly, evoked peak force did not recover in the 10 s between TF3 and TF3-post.

## 4. Discussion

The present study examined (i) resulting changes in measures of neuromuscular function during consecutive trials of isometric knee extension exercise, each of which terminated at progressively lower levels of voluntary muscle force development; (ii) the extent of recovery in neuromuscular activation, MVC force and evoked peak force within 5–10 s after task failure. 

Accordingly, the first important finding was that evoked peak forces were reduced more at task failure in the third trial than for the first and second trials, questioning whether a critical threshold in peripheral fatigue had been reached during the first two trials, and how peripheral fatigue influences neuromuscular activation and voluntary force at task failure.

The second important finding was that in the third trial neuromuscular activation and MVC force recovered substantially within 5–10 s of task failure without any recovery in evoked peak force. Thus, the present data indicate that central neuromuscular activation can change independently of any change in evoked peak force, our measure of peripheral fatigue.

### 4.1. Neuromuscular Activation and Peripheral Fatigue at Task Failure

In accordance with other studies using knee extensors [18,19], MVC force and evoked peak force decreased while RMS∙M^−1^ increased during exercise at constant force production during the first trial (Table 1 and Figure 2). However, since RMS∙M^−1^ during the last set of the first two trials and at the end of the third trial did not reach the higher values measured during the pre-exercise MVC, we suggest that task failure during exercise involving intermittent and sustained contractions is very likely influenced or caused by the inability to increase motoneuron drive. Additionally “similar” MVC and tetanus force at TF1, but different values at TF2 and more reduction in MVC compared with tetanus force at TF2 and TF3-post, suggest that the muscle was able to produce more force, which indicates that motoneuron drive was reduced. If group III and IV muscle afferents provide inhibitory feedback from locomotor muscles to the central nervous system [31] and exert a major influence to reduce neuromuscular activation, a lower RMS∙M^−1^ would be expected in the very final MVC of each trial. This was not the case in the first trial, in which RMS∙M^−1^ during the MVC at task failure was similar to the baseline value. This finding is not uncommon after exercise of a few minutes’ duration to task failure involving either isometric intermittent [15] or isometric sustained exercise [23], or after time trials of knee extension [14] or cycling [11]. However, a reduction in RMS∙M^−1^ for VM during the MVC at TF2, and for both VM and VL at TF3 corresponds with the reductions between MVC and tetanus force, indicating reduced motoneuron drive.

Reductions in evoked peak force at task failure were also lower at the end of the first two trials, indicating that there was less peripheral fatigue compared with the final trial. These results contrast with those from other studies that did not find any differences in the extent of peripheral fatigue in the knee extensors following isometric exercise to task failure in normoxia or hypoxia [32]; or in the same muscle group following self-paced 5 km cycling time trials comparing pre-fatigued and non-exercised (fresh) muscle groups [11], and following exercise involving cycling sprints comparing muscles either fresh or that had been pre-fatigued with electrical stimulation [33]. The reasons for these discrepancies are not clear; they are possibly related to differences in protocols, in the types of exercise used to produce different levels of peripheral fatigue, in the timing of measurements [34], or perhaps, in a combinations of these effects.

Studies using protocols more comparable to the present study found that the level of peripheral fatigue at task failure is higher following a pre-fatiguing isometric trial of the same muscle group using either intermittent [15] or sustained [23] contractions. Surprisingly, evoked peak forces for SS, PS100 and tetanus did not change significantly from task failure in the first compared with the second trial (Table 1), and thus, contrast with the results from the first experimental visit [15] and with the reduction in evoked peak force from the first to the third trial. We believe that this might have happened because two of the ten subjects seemed to lose motivation and “give up” exercise prematurely during the second trial (subjects 1 and 3, Figure 4). This led to an aberrant recovery in evoked peak forces. If the data for these two subjects were excluded, evoked peak forces were significantly lower (*p* < 0.001 for PS100) at the end of the second compared with the first trial. However, we could find no justification to remove their data from the final analysis.

Nevertheless, in comparison with the first experimental visit [15], eight of the ten subjects exercised for longer until reaching task failure on first trial of the second experimental visit (present study). However, the mean difference in time (2 min and 54 s, *p* = 0.07) or evoked peak force was not significantly different. A tendency of longer time to task failure is perhaps not surprising, as it has been shown [35] that a learning/familiarization effect can improve performance.

The findings from Froyd et al. [15] and Neyroud et al. [23], along with the present study, indicate that task failure during sustained or repeated unilateral isometric contractions of the knee extensors does not seem to be associated with a singular level of peripheral fatigue, and that peripheral fatigue increases when the pre-fatiguing isometric exercise utilizes the same muscle-groups as the subsequent trial. 

Since evoked peak force is reduced more after a final trial than a pre-fatiguing trial, it is relevant to question whether evoked peak force could be further reduced if experimental studies used more than just one trial per day to induce fatigue. For example, although direct comparisons are problematic, the reduction in potentiated evoked peak force at TF3 in the present study (−78% for SS, −80% for PS10 and −59% for PS100) is higher than in other studies; −48% for PS100 [23], −51% for SS [33], −58% for SS and −42% for PS100 [14], possibly due to the repeated exercise bouts and the length of the protocol in our trials.

Comparisons of peripheral fatigue between trials and the consequences of a critical peripheral fatigue threshold have been intensively discussed in recent years [36,37]. It has been proposed that in order to disprove the threshold concept an experimental investigation must cause the subjects to voluntarily surpass the task-specific threshold, i.e., more peripheral fatigue is necessary in the subsequent trial [36,38]. In contrast to Johnson et al. [13], by lowering the threshold for task failure in each subsequent trial, we detected more peripheral fatigue in the subsequent than in the prior trial. It is also hypothesized that the individual critical threshold in peripheral fatigue is task-specific [3,10]. Whether our study involving unilateral isometric intermittent and sustained contractions at varying target force fall within a single category of “task-specific” and hence whether it is a valid paradigm to question the threshold concept can be debated. Of note, rest breaks between contractions in the intermittent trials lasted five seconds, and similarly, the break between task failure at the end of the third trial and the following 5-s MVC was also 5 s. Notwithstanding these similarities, the decision whether the present study indeed compared similar tasks will depend on the definition of a task, including exercise intensity. Similar [39,40] and different [24,41] levels of peripheral fatigue between isometric trials of various intensities have also been reported in other studies, without specifically discussing the threshold concept. Nevertheless, in our understanding, increased peripheral fatigue in the final trial questions the role of peripheral fatigue on regulating exercise performance.

In addition, or alternatively, exercise performance may be influenced by a sensory tolerance limit [38]. A sensory tolerance limit may be reached with less peripheral fatigue when exercising with two compared with one leg, or when leg-cycling is pre-fatigued with arm-cycling [36,38]. The cause of exercise termination can be explained by a crossover of central fatigue from the other muscles [36]. Although we did not measure central fatigue, it is possible that central fatigue increased during such a long isometric trial [40]. If central fatigue developed, it did not prevent the increase in peripheral fatigue during the third trial.

### 4.2. Neuromuscular Activation at the End of the Third Trial and during the Immediate Post-Exercise Recovery Period

Despite a reduction in neuromuscular activation at task failure in the third trial, within 5 s of stopping exercise participants were again able voluntarily to generate neuromuscular activation similar to their pre-exercise values (Table 1 and Figure 4). Although the reduced muscle contractility at task failure can potentially recover within 10 s [42], evoked peak force was reduced by 78% for SS and 59% for PS100 compared with pre-exercise, and did not recover during the 10 s after task failure, even though neuromuscular activation nearly doubled during the same time (Table 1). The immediate recovery in neuromuscular activation without any recovery in evoked peak force indicate that the ability to continue exercising seems to be more dependent on central than peripheral factors [23], as indicated by the ~50% reduction in RMS∙M^−1^ at the end (TF3) of the final trial. The substantial decrease in both force and RMS∙M^−1^ at the end of the third trial are in accordance with trials of sustained MVC of the knee extensors [43] and ankle dorsiflexors [44]. 

To our knowledge, no other studies have documented such a substantial immediate recovery in MVC force and RMS∙M^−1^ simultaneously with no recovery in evoked peak force after a sustained MVC of the knee extensors. Nevertheless, this finding is in agreement with studies using sustained isometric contractions at 20%–35% MVC [22,23,24] and another in which subjects cycled to task failure, though showed an increase in voluntary power output during a final effort only a few seconds after task failure [8]. Immediate recovery in the capacity for force production seems to be present even when leg blood flow is occluded after task failure, preventing the clearance of muscle metabolites before measurements taken in the post-exercise recovery period [26,27]. 

The rapid increase in neuromuscular activation during the immediate post-exercise recovery period following fatiguing exercise might be explained by the recovery from increased intra-cortical inhibition [45] or by changes in intrinsic motoneuron properties [46]. However, while the silent period—generally associated with intra-cortical inhibition—recovers in 15 s after a 2-min sustained MVC, the responsiveness of spinal motoneurones requires ~90 s to return to control values [46]. Thus, the rapid recovery of the silent period after a fatiguing task when motoneurone responsiveness continues to be depressed is consistent with a cortical rather than a spinal origin for the rapid recovery of neuromuscular activation that we measured in the few seconds between task failure and the succeeding MVC in the third trial.

In line with the interpretations of Solianik et al. [43] and Kent-Braun [44], we suggest that both peripheral fatigue and reduced neuromuscular activation contribute to decreased voluntary force output during the trials. While MVC force at task failure (TF1, TF2 and TF3-post) was not substantially limited by sub-maximal muscle activation, the ability to sustain or increase neuromuscular activation was the decisive factor sustaining time to task failure during the trials, and this seemed to be relatively independent of the level of peripheral fatigue. Therefore, we suggest that it is important to separate the factors that may limit or determine performance during exercise from those that are measured during the MVC maneuver performed during the post-exercise recovery period. 

Importantly, the absence of a reduction in voluntary activation or neuromuscular activation [14,23,47] a few seconds after exercise cessation should not indicate that central mechanisms were not limiting or regulating performance during the preceeding exercise. In contrast, the rapid recovery of these neural controls to pre-exercise values immediately on the cessation of exercise indicates that they are likely the key controls regulating performance during exercise.

### 4.3. Methodological Considerations

Subjects knew beforehand that the experiment consisted of three trials and might therefore have terminated the first or second trial in a “submaximal” state of fatigue [48] despite strong encouragement. This is unfortunately an unavoidable feature of all studies involving consecutive trials of exercise [13,23,49]. However, prior knowledge of the expected duration of exercise trials does not necessarily influence subsequent performance [33].

## 5. Conclusions

Since task failure occurred at different levels of evoked peak forces in trials finishing at different levels of imposed force production, the present results question the concept that a critical peripheral fatigue threshold reduces neuromuscular activation. Accordingly, we concluded that the level of peripheral fatigue is not the major factor contributing or determining the onset of task failure. Furthermore, MVC force recovered substantially in the first 5–10 s after task failure in the third trial simultaneously with recovery in neuromuscular activation, but without any recovery in peripheral fatigue, providing evidence that immediate recovery in force production capacity at task failure is due to central and not peripheral mechanisms. Taken together, these findings suggest that central mechanisms should be considered the major contributors for task failure at the end of exercise.

## Figures and Tables

**Figure 1 sports-06-00156-f001:**
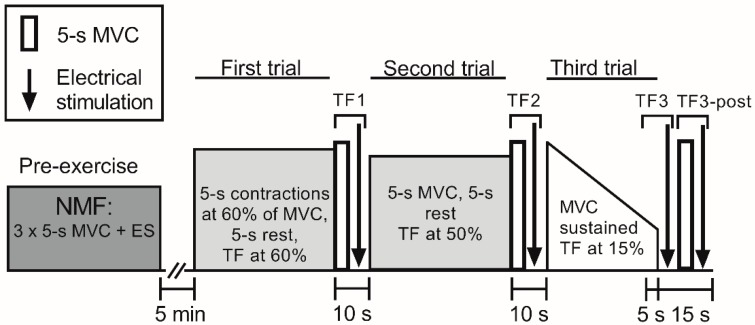
Diagram showing the protocol for the three trials. Neuromuscular function (NMF) (consisting of a 5-s maximal voluntary contraction (MVC)) followed immediately (<2 s) by electrical stimulation (ES) was assessed three times prior to the first trial, after every set during the first and second trials, and at task failure. First and second trials consisted of consecutive sets of 10 × 5-s isometric contractions followed by 5-s rest between contractions. In the first trial, the first nine contractions were performed at a target force set at 60% of the pre-exercise MVC, while the 10th contraction in each set was an MVC. Sets of contractions were repeated until task failure. In the second trial, sets of 10 × 5-s MVCs followed by 5-s rest between MVCs were performed until voluntary force dropped below 50% MVC. The third trial was a sustained MVC performed until force dropped below 15% MVC. Task failure of the first trial (TF1) and task failure of the second trial (TF2) comprised of a 5-s MVC followed by ES. Task failure of the third trial (TF3) comprised of the final 5 s of the third trial and the following ES. 5–10 s after the end of the third trial (TF3-post) comprised a 5-s MVC followed by ES. TF3-post started 5 s after cessation of the third trial.

**Figure 2 sports-06-00156-f002:**
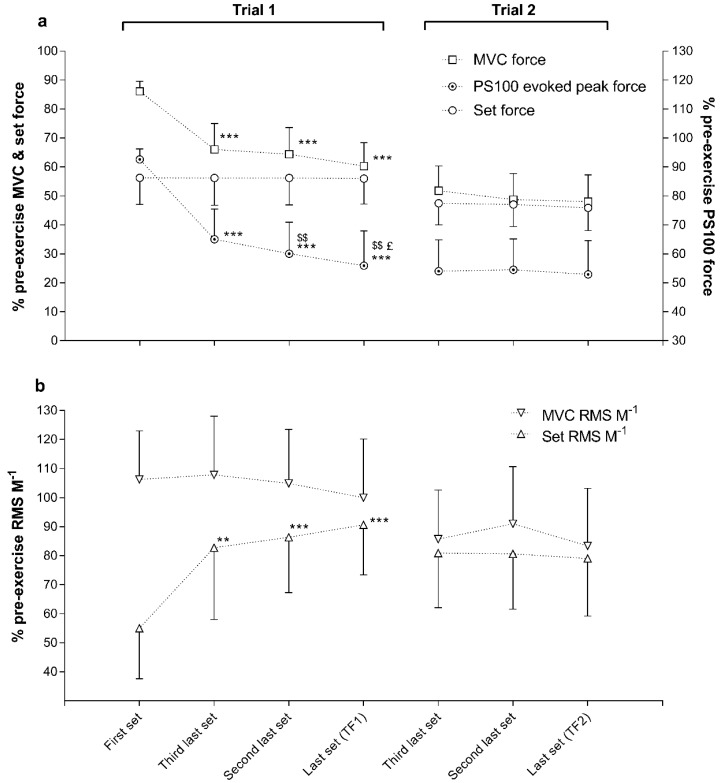
Responses in % of pre-exercise maximal voluntary contraction (MVC) force, set force, and evoked peak force for paired stimuli at 100 Hz (PS100) (panel (**a**)), and root mean square divided by M-wave peak to peak amplitude (RMS∙M^−1^) for MVC and set (panel (**b**)) of the first, third last, second last and the last set (TF1) for the first trial, and of the third last, second last and the last set (TF2) for the second trial. Data are means ± SD. Significant difference during the same trial, from first set: ** *p* < 0.01 and *** *p* < 0.001; from third last set: ^$$^
*p* < 0.01; from second last set: ^£^
*p* < 0.05.

**Figure 3 sports-06-00156-f003:**
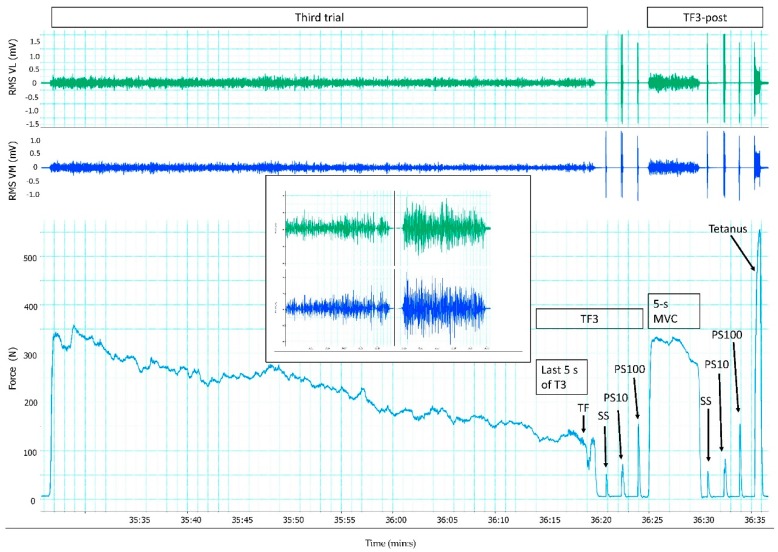
Responses in % of pre-exercise root mean square (RMS) for vastus lateralis (VL) and vastus medialis (VM), and force during the third trial (T3), task failure of the third trial (TF3) and TF3-post from one representative subject. Inserted is RMS for VL and VM during the last 5 s of T3 and the 5-s MVC of T3-post. T3 started more than 35 min into the protocol, and lasted approximately 52 s. Evoked responses of the single stimulus (SS), paired stimuli at 10 Hz (PS10), and paired stimuli at 100 Hz (PS100) were measured immediately the third trial and after a 5-s MVC. In addition, tetanus followed 1.5 s after PS100 during TF3-post.

**Figure 4 sports-06-00156-f004:**
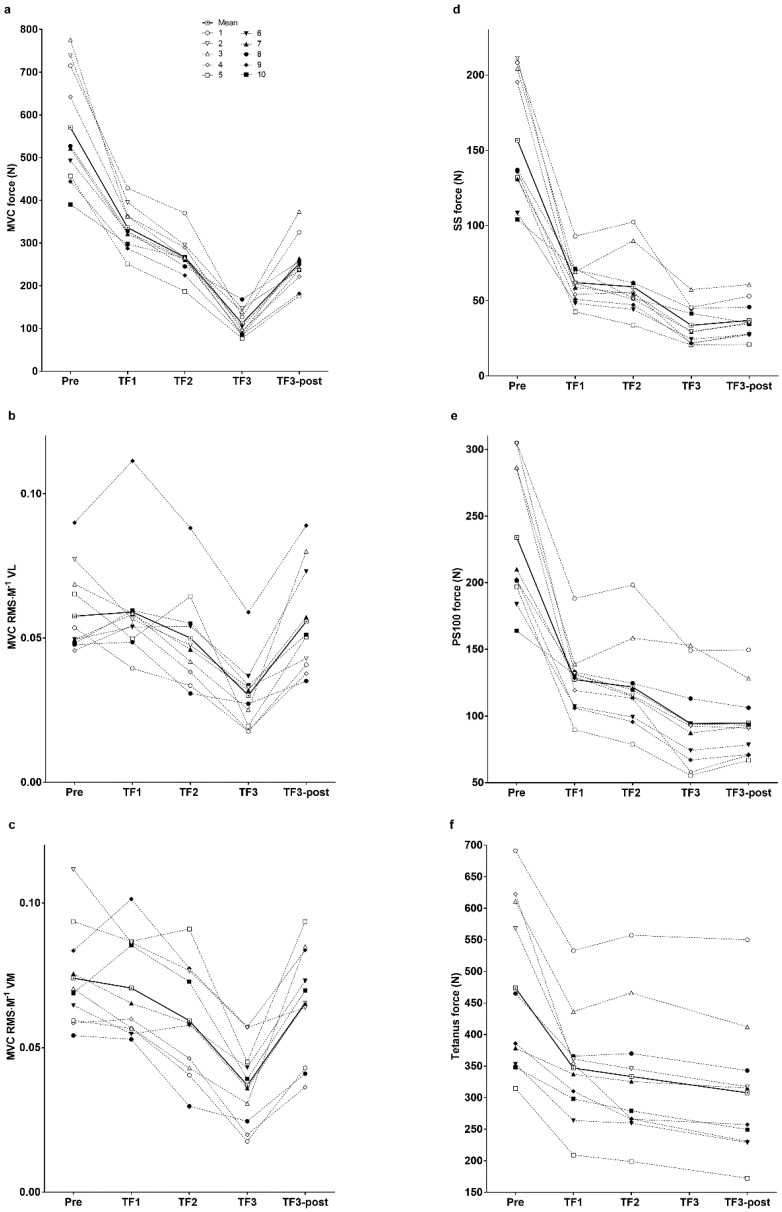
Individual and mean responses at pre-exercise, TF1, TF2, TF3 and TF3-post for MVC force (**a**), MVC RMS∙M^−1^ VL (**b**), MVC RMS∙M^−1^ VM (**c**), evoked peak force for SS (**d**), PS100 (**e**) and tetanus (**f**). Each subject is indicated with a symbol. Significant differences are presented in Table 1. MVC, maximal voluntary contraction; RMS∙M^−1^, root mean square divided by M-wave peak to peak amplitude; VL, vastus lateralis; VM, vastus medialis; SS, single stimulus; PS100, paired stimuli at 100 Hz; tetanus, 50 stimuli at 100 Hz; TF1, task failure of the first trial; TF2, task failure of the second trial; TF3, task failure of the third trial; TF3-post, task failure at 5–10 s after TF3.

**Table 1 sports-06-00156-t001:** Effects of knee extensor isometric exercise to task failure prior to exercise, at task failure of the three trials, and at 5–10 s after the third trial (Task Failure 3-post).

Parameter	Pre-Exercise	Task Failure 1	Task Failure 2	Task Failure 3	Task Failure 3-Post	Effect Size
MVC (N)	571 ± 137	336 ± 53 ***	267 ± 48 ***^,$$$^	112 ± 31 ***^,$$$,£££^	252 ± 60 ***^,$$,&&&^	η² = 0.914
Δ%		−40 ± 8	−52 ± 9 ^$$$^	−80 ± 5 ^$$$,£££^	−55 ± 9 ^$$,&&&^	η² = 0.921
MVC RMS∙M^−1^ VL	0.060 ± 0.015	0.059 ± 0.019	0.050 ± 0.017	0.030 ± 0.012 **^,$$$,££^	0.057 ± 0.019 ^&&^	η² = 0.611
Δ%		0 ± 21	−15 ± 21	−49 ± 17 ^$$$,£££^	−5 ± 27 ^&&&^	η² = 0.729
MVC RMS∙M^−1^ VM	0.074 ± 0.018	0.071 ± 0.018	0.059 ± 0.020 *^,$^	0.037 ± 0.014 ***^,$$$,££^	0.066 ± 0.020 ^&&^	η² = 0.708
Δ%		−4 ± 16	−21 ± 17 ^$^	−51 ± 13 ^$$$,£££^	−11 ± 22 ^&&&^	η² = 0.722
SS (N)	157 ± 43	62 ± 15 ***	59 ± 21 ***	34 ± 13 ***^,$$$,££^	37 ± 13 ***^,$$$,££^	η² = 0.898
Δ%		−58 ± 12	−62 ± 10	−78 ± 9 ^$$$,£££^	−76 ± 7 ^$$$,£££^	η² = 0.855
PS10	249 ± 66	95 ± 22 ***	83 ± 26 ***^,$^	48 ± 21 ***^,$$$,£££^	53 ± 17 ***^,$$$,£££^	η² = 0.909
Δ%		−60 ± 13	−66 ± 9 ^$^	−80 ± 9 ^$$$,£££^	−78 ± 7 ^$$$,£££^	η² = 0.873
PS100 (N)	234 ± 55	127 ± 26 ***	122 ± 34 ***	94 ± 35 ***^,$$,££^	95 ± 27 ***^,$$$,£££^	η² = 0.884
Δ%		−44 ± 12	−47 ± 12	−59 ± 13 ^$$,£££^	−59 ± 10 ^$$$,£££^	η² = 0.826
Tetanus (N)	474 ± 137	347 ± 90 **	334 ± 108 **	NA	308 ± 109 **^,$,££^	η² = 0.747
Δ%		−26 ± 10	−29 ± 13	NA	−35 ± 14 ^$$,£££^	η² = 0.618
PS10∙PS100^−1^	1.07 ± 0.08	0.75 ± 0.07 ***	0.67 ± 0.05 ***^,$$$^	0.49 ± 0.09 ***^,$$$,£££^	0.55 ± 0.07 ***^,$$$,£££,&^	η² = 0.968
Δ%		−30 ± 7	−37 ± 5 ^$$^	−54 ± 7 ^$$$,£££^	−48 ± 6 ^$$$,£££,&^	η² = 0.922
PPA *VL* (mV)	3.88 ± 0.67	3.66 ± 0.67	3.25 ± 0.80	3.09 ± 0.76 *	3.15 ± 0.72 *	η² = 0.572
Δ%		−5 ± 13	−16 ± 14	−21 ± 14 ^$^	−19 ± 13	η² = 0.503
PPA *VM* (mV)	3.74 ± 0.78	3.58 ± 0.75	3.32 ± 0.82	3.35 ± 0.91	3.39 ± 0.90	η² = 0.406
Δ%		−4 ± 11	−12 ± 12	−11 ± 14	−10 ± 13	η² = 0.396

Values are expressed in absolute units and as a percentage change from pre-exercise (Δ%). Values are expressed as means ± SD (*n* = 10). MVC, maximal voluntary contraction; RMS, root mean square; M, M-wave; VL, vastus lateralis; VM, vastus medialis; SS, single stimulus; PS10, paired stimuli at 10 Hz; PS100, paired stimuli at 100 Hz; Tetanus, 50 stimuli at 100 Hz; PPA, peak to peak amplitude of the M-wave; NA, not assessed. Significant difference compared with pre-exercise: * *p* < 0.05, ** *p* < 0.01, and *** *p* < 0.001; significant difference compared with Task Failure 1: ^$^
*p* < 0.05, ^$$^
*p* < 0.01, and ^$$$^
*p* < 0.001; significant difference compared with Task Failure 2: ^££^
*p* < 0.01, and ^£££^
*p* < 0.001; significant difference compared with Task Failure 3: ^&^
*p* < 0.05, ^&&^
*p* < 0.01, and ^&&&^
*p* < 0.001.

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
