# Peer review of "Neuromuscular Fatigue at Task Failure and During Immediate Recovery after Isometric Knee Extension Trials"

_sports, 2018, doi:10.3390/sports6040156_

Round 1
Reviewer 1 Report
General Comment
The aim of the manuscript sports- 385144 was to evaluate if the level of peripheral fatigue would differ when three consecutive exercise trials were completed to task failure, and whether there would be delayed recovery in maximal voluntary contraction (MVC) force, neuromuscular activation and peripheral fatigue following task failure. This paper is well written and with strong impact in our current knowledge on the exercise induced fatigue mechanism, with a very specific and well conducted experimental design. However, they are some points that must be addressed
Specific comments.
Introduction
Lines 43-44. A more detailed report about the physiological mechanism underlying peripheral fatigue is needed. So do the refs
Lines 57-59. Please re-phrase because it is very difficult to be followed
A better description of the critical peripheral fatigue threshold, and the physiological background of it are necessary somewhere in the introduction section
Materials and Methods
Please provide the power analysis results for the present study
Subjects. Given that in the present study, participants were well-trained in resistance and endurance training, their neuromuscular adaptations and the mechanism that leads to fatigue is different compared to novice ones, authors should discuss it in the discussion section. In addition, authors should discuss the possible difference in both the rates, mechanism and the recovery between endurance trained and resistance trained participants. Finally, due to the fact that in the present study, women were also participated, authors do they control/know the time of their menstruation, while they should discuss about the effect of the estrogen on neuromuscular fatigue and or exercise induced fatigue or muscle damage. They were any differences between men and women?
Subjects. Please provide a detailed description about the trainings that these participants were follow. Do authors control the trainings that their participants performed at least 1 week prior to the experimental setting. It is well knowed that after a heavy or intense training the neuromuscular fatigue is present until 78h after the training session
Experimental design. According to the design of the present sudy it seems that the intense effort should at least compromise the capability of maximum isometric effort. Thus, it will be very interesting if authors provide data about the initial forces generated in each trial as well as the rate of decrease of these forces, especially this is very important to the last trial, were the end was determinated as the 15% of the pre-test force.
Authors should also explain why they choose this experimental design, based on physiological mechanism
Procedures, Data collection and data analysis. A better and a more detailed description about the procedures of EMG evaluations are need here. Especially for the calculation of EMG signal, RMS, M-Wave and EMG data. Please provide the ICCs. It will be very helpful if authors provide a picture from the setting of EMG evaluations and analyses
Statistical analyses. Please provide the effect size in each comparison of each parameter. In which parameters the assumption of sphericity was or not violated?
Table 1. the symbol that have been used for the report of the difference are very confused. Please change it in a manner of a more reader friendly table
Fig 4. According to these figure it seems that they are participants which could be characterized as responders and others as non responders. It will be very interesting if authors explore the difference between responders and non-responders. What were their characteristics and how they impair the individual responses?
Discussion
The discussion section is very difficult to be followed. Authors should re-writing it, considering the fact that not all readers are familiar either with the terms or the physiological background. This should lead to a better and more extensive spread of the message that his study wants to provide. Finally, authors should consider to discuss their results, and the possible physiological mechanism underlying them, in a more time and intensity of the exercise manner.
Please start your discussion with the main findings of the present study
Authors should also discuss in extensive manner, the physiological mechanism underlying their results
Lines 315-317. A feedback inhibitory of group III and IV muscle afferents, has been reported previously but in amateur patricipants. No in well trained ones. Thus authors should discuss the possible training specific adaptations that are expected in either resistance or endurance trained participants, as well as their possible effect on the group III and IV muscle afferents feedback mechanism and on RMS∙M-1. They were any difference between endurance and resistance trained participants?
Author Response
Point 1: The aim of the manuscript sports- 385144 was to evaluate if the level of peripheral fatigue would differ when three consecutive exercise trials were completed to task failure, and whether there would be delayed recovery in maximal voluntary contraction (MVC) force, neuromuscular activation and peripheral fatigue following task failure. This paper is well written and with strong impact in our current knowledge on the exercise induced fatigue mechanism, with a very specific and well conducted experimental design. However, they are some points that must be address
Response 1: We would like to thank the reviewer for the kind words.
Introduction
Point 2: Lines 43-44. A more detailed report about the physiological mechanism underlying peripheral fatigue is needed. So do the refs
Response 2: We believe that our description is sufficient. As we are not measuring these mechanisms, we believe a more expansive review would be misplaced. We have added a reference (#3) to the sentence.
Point 3: Lines 57-59. Please re-phrase because it is very difficult to be followed
Response 3: The sentence has been modified (line 59).
Point 4: A better description of the critical peripheral fatigue threshold, and the physiological background of it are necessary somewhere in the introduction section
Response 4: We believe that the present description is sufficient (see lines 50-62), as it is very close to that used by the original proponents. A full review of the theory is beyond the scope of the present paper.
Materials and Methods
Point 5: Please provide the power analysis results for the present study
Response 5: The study was done on a convenience sample, which for logistical reasons could not be expanded. However, the sample size and variability of measurements (not reported) is well within what is typically reported in literature for studies on neuromuscular fatigue.
Point 6: Subjects. Given that in the present study, participants were well-trained in resistance and endurance training, their neuromuscular adaptations and the mechanism that leads to fatigue is different compared to novice ones, authors should discuss it in the discussion section. In addition, authors should discuss the possible difference in both the rates, mechanism and the recovery between endurance trained and resistance trained participants. Finally, due to the fact that in the present study, women were also participated, authors do they control/know the time of their menstruation, while they should discuss about the effect of the estrogen on neuromuscular fatigue and or exercise induced fatigue or muscle damage. They were any differences between men and women?
Response 6: The reviewer is correct in that trained and untrained participants might have differences in their response to fatiguing trials. However, we disagree that we should discuss the difference between novice and trained subjects, as that is not pertinent to the present investigation and we could do no more than speculate. Time in the menstrual cycle was not controlled for, but given the general lack of effect of the menstrual cycle on performance (except for deleterious effects of cramping etc, which was not the case in our study), we believe that would little effect of it in our findings. Differences between men and women was not the aim of the study.
Point 7: Subjects. Please provide a detailed description about the trainings that these participants were follow. Do authors control the trainings that their participants performed at least 1 week prior to the experimental setting. It is well knowed that after a heavy or intense training the neuromuscular fatigue is present until 78h after the training session
Response 7: No registry of training data was performed. Subjects were instructed not to perform any high intensity or high effort activity on the day before and on the day of testing (before the test). Subjects performed better on pre-exercise MVC during the second compared to first experimental visit, and their force response to electrical stimulation was similar pre-exercise between the two experimental days. The data from the first experimental visit is already published (Froyd, C.; Beltrami, F.; Millet, G.; Noakes, T. No critical peripheral fatigue threshold during intermittent isometric time to task failure test with the knee extensors. Frontiers in Physiology 2016, 7, doi:https://doi.org/10.3389/fphys.2016.00627.)
Point 8: Experimental design. According to the design of the present sudy it seems that the intense effort should at least compromise the capability of maximum isometric effort. Thus, it will be very interesting if authors provide data about the initial forces generated in each trial as well as the rate of decrease of these forces, especially this is very important to the last trial, were the end was determinated as the 15% of the pre-test force.
Response 8: If we understand the reviewer correctly, he/she asks for the MVC force in the start of each trial. We have presented pre-exercise values before the first trial (Table 1) and MVC force after the first set of trial 1 and trial 2 (Figure 2a). In figure 3, we have presented absolute force from the start of the first trial for one representative subject. In average this force was about the same as MVC force at task failure 2 (-52%) (Table 1). We think the reader can find these results in the manuscript.
Point 9: Authors should also explain why they choose this experimental design, based on physiological mechanism
Response 9: The rationale for the study is explained in the introduction (lines 87-94).
Point 10: Procedures, Data collection and data analysis. A better and a more detailed description about the procedures of EMG evaluations are need here. Especially for the calculation of EMG signal, RMS, M-Wave and EMG data. Please provide the ICCs. It will be very helpful if authors provide a picture from the setting of EMG evaluations and analyses
Response 10: We disagree with the reviewer. Our description of data collection and data analysis follow standard procedures for this type of methods and equipment, according the latest guidelines for EMG collection and interpretation.
Point 11: Statistical analyses. Please provide the effect size in each comparison of each parameter. In which parameters the assumption of sphericity was or not violated?
Response 11: We chose not to report individually if sphericity had been violated, as is normal in the literature. If the reviewer insists, we can add the eta-squared values to the results, however we feel like most of the readership will not be able to extract meaningful information from them.
Point 12: Table 1. the symbol that have been used for the report of the difference are very confused. Please change it in a manner of a more reader friendly table
Response 12: Our symbols and way of use are common in literature. The symbols are repeated to indicate significance level and different symbols are used to indicate different levels of comparisons. As there are several time-points, there will be many different symbols no matter how one tries to accommodate them. If the reviewer has a concrete suggestion on how to improve clarity without losing information, we would happy to consider it.
Point 13: Fig 4. According to these figure it seems that they are participants which could be characterized as responders and others as non responders. It will be very interesting if authors explore the difference between responders and non-responders. What were their characteristics and how they impair the individual responses?
Response 13: There was no intervention to be a “responder” or not. If the reviewer is considering “fatigures” or “non-fatiguers”, we contend that our data does not suggest figure. Furthermore, splitting participant on visual inspection will lead to a return to the mean bias, where participants with more extreme values by rule tend to return to more common values on subsequent examination, a consequence of measurement error and variability that has little to do with being a “fatiguer” or “non-fatiguer”. Doing a more in-depth investigation of such possible phenotypes would be interesting, but would require far more individuals who would need to be proven to be fatigue-prone or fatigue-resistant, and is beyond the scope of our study.
Discussion
Point 14: The discussion section is very difficult to be followed. Authors should re-writing it, considering the fact that not all readers are familiar either with the terms or the physiological background. This should lead to a better and more extensive spread of the message that his study wants to provide. Finally, authors should consider to discuss their results, and the possible physiological mechanism underlying them, in a more time and intensity of the exercise manner.
Response 14: The manuscript is submitted to a special issue “Neuromuscular function …”, and the readers are expected to be somewhat familiar with neuromuscular function testing and fatigue. We consider that suggestion that “author should re-writing it” to be somewhat baffling.
Point 15: Please start your discussion with the main findings of the present study
Response 15: Thank you for this advice. That is exactly how we started the discussion (lines 298-305).
Point 16: Authors should also discuss in extensive manner, the physiological mechanism underlying their results
Response 16: We have discussed physiological mechanisms. Please let us know which mechanisms and line numbers.
Point 17: Lines 315-317. A feedback inhibitory of group III and IV muscle afferents, has been reported previously but in amateur patricipants. No in well trained ones. Thus authors should discuss the possible training specific adaptations that are expected in either resistance or endurance trained participants, as well as their possible effect on the group III and IV muscle afferents feedback mechanism and on RMS∙M-1. They were any difference between endurance and resistance trained participants?
Response 17: This is not the aim of the study, and we do not have data to discuss it.
Reviewer 2 Report
This manuscript reports research on Neuromuscular fatigue at task failure and during immediate recovery after isometric knee extension trials. There are several studies which have reported a similar analysis and hence, authors need to address some of the major concerns before the paper is considered for publication.
There are several reported studies which investigated fatigue and MVC assessments using BSS/ICA, NMF and other methods. The following related papers will be useful:
Bryanton, Megan A., and Martin Bilodeau. "The influence of knee extensor fatigue on lower extremity muscle activity during chair rise in young and older adults." European journal of applied physiology (2018): 1-11.
Arjunan et al,. "Computation and evaluation of features of surface electromyogram to identify the force of muscle contraction and muscle fatigue." BioMed research international 2014 (2014).
Mira, J., et al. "Effects of endurance training on neuromuscular fatigue in healthy active men. Part I: Strength loss and muscle fatigue." European journal of applied physiology 118.11 (2018): 2281-2293.
Naik et al,. "Testing of motor unit synchronization model for localized muscle fatigue." In Engineering in Medicine and Biology Society, 2009. EMBC 2009. Annual International Conference of the IEEE, pp. 360-363. IEEE, 2009.
Souron, R., Nosaka, K., & Jubeau, M. (2018). Changes in central and peripheral neuromuscular fatigue indices after concentric versus eccentric contractions of the knee extensors. European journal of applied physiology, 118(4), 805-816.
Kumar et al, 2011. Measuring increase in synchronization to identify muscle endurance limit. IEEE transactions on neural systems and rehabilitation engineering, 19(5), pp.578-587.
Specific comments to each section
Introduction section needs to be further improved with more explanation and literature review.
Authors used EMG data which are collected using a large number of sensors and hence they need to explain how they have addressed cross-talk related to EMG data (because EMG sensors are connected in close proximity).
Authors need to explain how or what level of force was used as an indicator of MVC. It is stated in the methods that 60% of force etc, but it is not clear. Please elaborate on that.
This study uses only 10 subjects and hence authors need to explain how they handled statistical analysis (parametric or non-parametric etc).
It is not clear how the NMF features or methods are used for the analysis, please explain in detail.
The classification accuracy needs to be supported with specificity, sensitivity and ROC
Author Response
Point 1: This manuscript reports research on Neuromuscular fatigue at task failure and during immediate recovery after isometric knee extension trials. There are several studies which have reported a similar analysis and hence, authors need to address some of the major concerns before the paper is considered for publication.
There are several reported studies which investigated fatigue and MVC assessments using BSS/ICA, NMF and other methods. The following related papers will be useful:
Response 1: The reviewer highlights several papers that he/she thinks should be considered:
Mira et al. (2018), Bryanton & Bilodeau (2018), and Souron et al. (2018) are interesting papers, but not directly relevant to our manuscript.
Souron et al. (2018): It would be interesting to discuss e.g post-exercise isometric MVC RMS/M (after concentric or eccentric exercise), but this was not reported. They reported no decrease in RMS/M for the bouts of exercise, but since this was not isometric exercise, we do not want to compare with our study. Neither did they report any immediate recovery in MVC torque or RMS/M.
Naik et al., 2009, Kumar et al., 2011, Arjunan et al., 2014: all of them use EMG to assess muscle fatigue. They are not using knee extensors or measuring peripheral fatigue with electrical stimulation. As we understand, they are discussing the limitations of using EMG to assess local muscle fatigue. However, we used RMS/M as a measure of neuromuscular activation. We agree that RMS/M has limitations, as we have discussed. However, despite these limitations, we think it is interesting to use RMS/M when discussing the recovery in MVC force without no recovery in peripheral fatigue (responses to electrical stimulation).
Specific comments to each section
Point 2: Introduction section needs to be further improved with more explanation and literature review.
Response 2: Our introduction is in line with the field, which is meant for readers familiar with neuromuscular function testing and fatigue. Excellent reviews of literature are available for those needing introductory content.
Point 3: Authors used EMG data which are collected using a large number of sensors and hence they need to explain how they have addressed cross-talk related to EMG data (because EMG sensors are connected in close proximity).
Response 3: We have used only two sensors (vastus lateralis and and vastus medialis, and we have as reported followed the recommendations from SENIAM, in addition to recommendations from Delsys.
Point 4: Authors need to explain how or what level of force was used as an indicator of MVC. It is stated in the methods that 60% of force etc, but it is not clear. Please elaborate on that.
Response 4: MVC is a maximal voluntary maneuver, and the maximal force produced is taken as 100%, from which the target forces are derived. This is explained in methods.
Point 5: This study uses only 10 subjects and hence authors need to explain how they handled statistical analysis (parametric or non-parametric etc).
Response 5: Parametric statistics were used, which are robust enough to be used with our sample size, especially on dependent (matched pairs or repeat tests), as in our case.
Point 6: It is not clear how the NMF features or methods are used for the analysis, please explain in detail.
Response 6: The method section clearly explains how NMF is measured with the different stimulations.
Point 7: The classification accuracy needs to be supported with specificity, sensitivity and ROC
Response 7: We disagree with the reviewer. We believe the analysis performed are adequate and in line with the field. The current study was not designed to provide figures of specificity and sensitivity.